# Particles Morphology Impact on Cytotoxicity, Hemolytic Activity and Sorption Properties of Porous Aluminosilicates of Kaolinite Group

**DOI:** 10.3390/nano12152559

**Published:** 2022-07-26

**Authors:** Olga Yu. Golubeva, Yulia A. Alikina, Elena Yu. Brazovskaya

**Affiliations:** Laboratory of Silicate Sorbents Chemistry, Institute of Silicate Chemistry of Russian Academy of Sciences, Adm. Makarova emb., 2, 199034 St. Petersburg, Russia; morozowa_u_a@mail.ru (Y.A.A.); brazovskaya.ics@gmail.com (E.Y.B.)

**Keywords:** kaolinite, nanoparticles, synthesis, morphology, cytotoxicity, hemolytic ability

## Abstract

A comparative study of the properties of aluminosilicates of the kaolinite (Al_2_Si_2_O_5_(OH)_4_∙nH_2_O) group with different particles morphology has been carried out. Under conditions of directed hydrothermal synthesis, kaolinite nanoparticles with spherical, sponge, and platy morphologies were obtained. Raw nanotubular halloysite was used as particles with tubular morphology. The samples were studied by X-ray diffraction, SEM, solid-state NMR, low-temperature nitrogen adsorption, and the dependence of the zeta potential of the samples on the pH of the medium was defined. The sorption capacity with respect to cationic dye methylene blue in aqueous solutions was studied. It was found that sorption capacity depends on particles morphology and decreases in the series spheres-sponges-tubes-plates. The Langmuir, Freundlich, and Temkin models describe experimental methylene blue adsorption isotherms on aluminosilicates of the kaolinite subgroup with different particles morphology. To process the kinetic data, pseudo-first order and pseudo-second order were used. For the first time, studies of the dependence of hemolytic activity and cytotoxicity of aluminosilicate nanoparticles on their morphology were carried out. It was found that aluminosilicate nanosponges and spherical particles are not toxic to human erythrocytes and do not cause their destruction at sample concentrations from 0.1 to 1 mg/g. Based on the results of the MTT test, the concentration value that causes 50% inhibition of cell population growth (IC_50_, mg/mL) was calculated. For nanotubes, this value turned out to be the smallest—0.33 mg/mL. For samples with platy, spherical and nanosponge morphology, the IC_50_ values were 1.55, 2.68, and 4.69 mg/mL, respectively.

## 1. Introduction

It is known that the shape and morphology of the particles can affect the physicochemical properties of composite materials based on them. Thus, the shape of filler particles affects the mechanical, electrical, gas transport, barrier property, thermal (thermal stability and flame retardancy) and other properties of polymer-inorganic nanocomposites [1,2,3,4,5]. The shape of filler particles and their aspect ratio (the length-to-diameter (l/d) ratio) also have an effect on the mechanical properties of ceramic composite materials and cement mortars [6,7,8,9,10]. At the same time, the influence of the shape and morphology of particles on their own physicochemical characteristics and biological activity is still poorly understood.

It is generally accepted that the properties of nanoparticles and nanomaterials are determined only by surface properties, chemical composition, particle size and their porous-textural characteristics. However, it appears that nanomaterials shape can influence systemic toxicity, biodestruction and blood circulation time. [11]. Nanoparticle morphology is one of the main factors that modulate the rate and mechanism of cellular uptake, as well as the intracellular transport [12,13,14,15]. Using copolymer nanoparticles as an example, it was shown that nanoparticle morphology is an important factor in the design of drug delivery systems due to its impact on behavior such as cell uptake, drug release kinetics and diffusion in complex biological media [16]. Studies of CeO_2_–MnO_x_ mixed oxides have shown that the particle morphology also affects their catalytic activity [17]. Furthermore, shape has also been used to modulate the antimicrobial effect of silver nanoparticles [18]. Besides, the morphology control of the particles demonstrates the importance of the ratio of the different crystal facet areas, which have a different dissolution rate in solution [19]. 

Aluminosilicates of the kaolinite subgroup (kaolinite and halloysite) are interesting in that they can form particles with different morphologies—platy, spherical, tubular, cylindrical, spongy, etc. [20,21,22,23]. Having the same structure and the same chemical composition, and corresponding to the general chemical formula Al_2_Si_2_O_5_(OH)_4_, such particles may exhibit different properties [24]. Unlike natural minerals, which in most cases are a mixture of particles of different morphology, under the conditions of directed hydrothermal synthesis, synthetic aluminosilicates with a given particle morphology can be obtained. Thus, aluminosilicates of the kaolinite subgroup can be used as ideal model objects for studying the influence of the morphology and shape of particles on their physicochemical properties and biological activity. Such information can open up new possibilities for the obtaining of the effective materials for medicine and ecology. 

The aim of the work was to study and establish the regularities of the possible influence of the morphology and shape of particles of kaolinite group aluminosilicates on their sorption capacity, hemolytic activity and cytotoxicity. The results of the study will make it possible to suggest assumptions about the potential of using particles with a certain morphology in various fields; in particular, in medicine.

## 2. Materials and Methods

### 2.1. Reagents

The following reagents were used for the synthesis and analysis of the samples: tetraethoxysilane TEOS ((C_2_H_5_O)_4_Si, special purity grade, ≥99.0%), aluminum nitrate Al(NO_3_)_3_∙9H_2_O (reagent grade, ≥97.0%), ammonia NH_4_OH (25 wt % NH_3_ in H_2_O), ethanol C_2_H_5_OH (96 wt %), methylene blue C_16_H_18_N_3_SCl (chemically pure grade), hydrochloric acid HCl (35–38 wt %), sodium hydroxide solution (50 wt % in water), raw halloysite nanotubes . All reagents used were from Sigma Aldrich (Burlington, MA, USA).

### 2.2. Synthesis

Samples of aluminosilicates were synthesized under hydrothermal conditions from dried gels of compositions corresponding to the chemical formula of kaolinite Al_2_Si_2_O_5_(OH)_4._ Gels were prepared using tetraethoxysilane, aluminum nitrate, ammonia, and ethanol according to the procedure detailed in [23,24].

The dried gels were placed in steel autoclaves with platinum or Teflon crucibles. The crucible material was determined by the synthesis temperature. Previously determined the optimal conditions for the crystallization of aluminosilicates with a given morphology (Golubeva et al., 2020, 2021) are given in Table 1. Solutions were preliminarily prepared with specified pH values (2.6, 7, and 12) by using solutions of HCl and NaOH. The pH values were measured using digital pH meter MEGEON PH 17206, the measurement accuracy is 0.1 pH. For the hydrothermal treatment, 1 g of dried gel was placed in a crucible and filled with 35 mL of the corresponding solution. The resulting product was washed with water and dried.

### 2.3. Characterization

X-ray phase analysis of the samples was carried out using a powder diffractometer Rigaku Corporation (Tokyo, Japan), SmartLab 3 (CuKα—radiation, operating mode—40 kV/40 mA; semiconductor point detector (0D)—linear (1D), θ-θ geometry, measurement range 2ϑ = 5–70° (step 2θ = 0.01°)).

Chemical analysis of the samples for the content of Si, and Al was carried out by the gravimetric method using quinolate of a silicon-molybdenum complex and by complexometric titration at pH 5. The content of H_2_O was estimated from the weight loss upon calcination of the sample at 1000–1100 °C

The textural parameters of the materials were determined by the method of low-temperature adsorption–desorption of nitrogen. Isotherms were obtained using a Quantachrome NOVA 1200e instrument (Quantachrome Instruments, Boynton Beach, FL, USA). Degassing was carried out at 300 °C for 12 h. 

The morphology of the samples was studied by scanning electron microscopy (SEM) by using a Carl Zeiss Merlin instrument (Oberkochen, Germany) with a field emission cathode. Powders of the samples were planted directly on conductive carbon tape without additional processing.

The electrokinetic (zeta) potential of the samples was determined using the particle size and zeta potential analyzer NaniBrook 90 PlusZeta (Brookehaven Instruments Corporation, Holtsville, NY, USA). The samples were a suspension obtained by dispersing 50 mg of sample in 20 mL of deionized water. Before measurements, the suspension was subjected to low power (50 W) ultrasonication for two minutes on an ultrasonic processor UP50H.

The pH values of the medium were varied in the range from 2 to 10 using HCl and NaOH (50 wt %) solutions. The pH value was measured using a MEGEON PH 17206 digital pH meter (OOO Megeon, Moscow region, Korolev, Russia); the measurement accuracy was ±0.1 pH. 

High-resolution NMR spectra in solids were obtained by rotating at a “magic” angle at room temperature on an AVANCE II-500WB spectrometer (Bruker Corporation, Billerica, MA, USA). Operating frequency for 29Si: 99.35 MHz, 27Al: 130.32 MHz. The spectra were recorded using a single-pulse excitation technique, pulse duration: 3 μs (π/4) and 0.6 μs (π/12) with a delay of 6 s and 0.5 s for ^29^Si and ^27^Al, respectively. The samples were packed in zirconium 4 mm rotors and rotated at a frequency of 13 kHz. Chemical shifts are given in ppm relative to TMS (tetramethylsilane).

The adsorption capacity of the samples was determined with respect to the cationic organic dye—methylene blue (MB). The study of the equilibrium adsorption of MB was carried out at the concentration of MB in the range from 10 to 400 mg/L. To do this, 20 mg of the sample was dispersed in 20 mL of an aqueous dye solution. The experiments were carried out in a static mode at room temperature in closed glass bottles with a volume of 50 mL while stirring for 120 min, which corresponded to the moment when adsorption equilibrium was established. The samples were filtered and the concentration of dyes in the filtrate was determined as the arithmetic mean of three measurements. The concentration of MB was determined using UV absorption spectroscopy (LEKISS2109UV spectrophotometer, LEKI Instruments, Helsinki, Finland) by the optical density at a wavelength of 246 nm [25]. 

The capacity of the sorbent, mg/g (the amount of adsorbed substance) was determined by the following Formula (1):(1)$$X=\frac{\left(C_{i}-C_{f}\right)\;\cdot \;V_{s}}{m_{s}},$$
where *C_i_* is the initial concentration of MB solution, g/L; *C_f_*—final concentration after sorption, g/L; *V_s_* is the volume of MB solution, L; *m_s_*—weight of the sorbent sample, g.

Comparison of the sorption capacity of the samples with known sorbents was carried out using activated carbon (brand DARCO^®^, Fluka, M = 12.01 g/mol, analytical grade).

The study of the adsorption kinetics of MB was carried out in a solution of dye with a concentration of 0.1 g/L with regular sampling (2 mL) at certain time intervals. After centrifugation of the sample for 15 min at 5000 rpm, the concentration of MB in the supernatant solution was determined spectrophotometrically from the absorption maximum using a calibration curve.

Langmuir, Freundlich and Temkin isotherm models were used to fit the data obtained [26,27,28]. The applicability of the isotherm equations to the adsorption isotherm was compared by the correlation coefficients, R-squared value (R^2^). The parameters of the adsorption equations were calculated by the method of nonlinear regression using the OriginPro 8 program. To process the kinetic data, pseudo-first-order (PFO) and pseudo-second-order (PSO) [29,30] adsorption models, as well as diffusion kinetic models [31,32] were used. A more detailed description of the models used is given in Appendix B.

In order to assess the toxicity of samples in relation to blood cells, the process of destruction of erythrocytes with the release of hemoglobin into the environment (hemolytic activity) was studied. A standard procedure [33,34] was used to determine the hemolytic activity. Cytotoxicity of the samples was assessed using standard MTT assay protocols, which makes it possible to evaluate the total activity of mitochondrial respiratory enzymes [35]. Cultured cells of tumor origin, human histiocytic lymphoma cells (U937), were used in the experiments. A detailed description of the methods for studying hemolytic activity and cytotoxicity of samples is given in Appendix C and Appendix D.

## 3. Results and Discussion

The X-ray diffraction patterns of samples are shown in Figure 1. All synthesized samples gave characteristic X-ray peaks at 7.14 and 3.57 Å for a kaolinite mineral [36]. The dominant peaks were found at 2θ = 11.8–12.0°, 20.07°, 24.9°, 35.1°, 38.1°, 55.08° and 62.2° which correspond to the reflection planes (001), (100), (002), (110), (003), (210) and (300), respectively.

In addition to the diffraction patterns of synthetic samples with different morphology, Figure 1 also shows the diffraction pattern of a sample of raw halloysite with nanotubular morphology. As can be seen from the diffraction pattern, the position of the main reflections corresponds to the reflection planes of halloysite (7 Å) [37,38]. 

Previous studies of the influence of synthesis conditions on the crystallization of aluminosilicates of the kaolinite subgroup with a given morphology [23,24] made it possible to determine the optimal conditions for obtaining samples with one predominant morphology. Thus, it was found that an alkaline environment (pH 12) favors the formation of particles with a predominantly spherical morphology, an acidic environment and a synthesis duration of 1 to 3 days (pH 2.6) favors the crystallization of particles with a nanosponge morphology. Hydrothermal crystallization in a neutral medium makes it possible to obtain the entire spectrum of particle morphologies—nanosponge (at 200 °C), spherical (220–300 °C) and platy (300–350 °C). Particles with tubular morphology can also be formed in neutral and alkaline media (see Appendix A); however, samples for which tubular morphology would be predominant have not been obtained. Therefore, in order to compare the properties of aluminosilicate samples with different morphology, a sample of raw halloysite was used as a sample with nanotubular morphology (Sigma-Aldrich, Product of Applied Minerals, Saint Louis, MO 63103, USA).

Table 2 presents the results of the chemical analysis of all samples, from which it follows that the samples under study are aluminosilicates with a very similar content of both silicon oxide and aluminum oxide.

Figure 2 shows the results of the study of samples using scanning electron microscopy. As can be seen from the presented micrographs, the studied samples are particles with spherical (Figure 2a), plate (Figure 2b), sponge (Figure 2c) and tubular morphology. Particles with spherical morphology have an average diameter of about 300 nm. Samples with nanosponge morphology are formed by aluminosilicate layers with a thickness of about 24–27 nm. Platy particles have a thickness of about 100 nm. The nanotubular sample is approximately 700 nm long and 60 nm in diameter.

Differences in the morphology of the particles should lead to differences in their porous-textural characteristics and specific surface values, which is confirmed by the results of the study. Figure 3 shows low-temperature nitrogen adsorption curves for the studied samples of different morphology. It can be seen from the figure that the samples with nanotubular and platy morphologies are characterized by similar shapes of hysteresis loops. Such a shape of isotherm can be attributed to type 4 isotherms according to the IUPAC classification (Figure 3b,d) and indicates the presence of both micro- and mesopores. In this case, the hysteresis loop can be attributed to the *H3* type and indicates the presence of aggregates of platy particles that form slit-like pores. The shape of the hysteresis loop of the synthesized aluminosilicate with a spherical and sponge morphology of particles can also be attributed to type 4 isotherms (Figure 3a,c), but the hysteresis loop is characteristic of the *H2* type. The samples differ considerably in their specific surface area, which increases from 47 to 500 m^2^/g depending on the particle morphology. 

Figure 4 shows the ^29^Si and ^27^Al MAS NMR spectra of samples. The bulk of aluminum in all samples is in an octahedral environment. This is evidenced by the dominant peak in the region of 5 ppm (Al^VI^) [39,40]. At the same time, part of aluminum in all samples is in a tetrahedral environment, which is confirmed by the presence of peaks in the region of 83 ppm (nanosponges, spheres, tubes), 70–74 ppm (spheres, plates) and 60 ppm (sponges) designated as Al^IV^_b_, Al^IV^_c_, Al^IV^_d_, respectively. The area from 60 to 90 ppm corresponds to 4-coordinated Al sites [40]. The presence of several signals in the region of 4-coordinated Al indicates a different position of aluminum in the tetrahedral layers and differences in its environment with silicon [41]. Thus, according to (Breen et al., 1995) the Al(IV) signal at ca. 71 ppm is due to the substitution of Al^3+^ for Si^4+^ in tetrahedral sheets while the one at ca. 55–60 ppm corresponds to Al in the three-dimensional silica framework. This agrees with the results of studying the samples by electron microscopy, which prove the presence of a three-dimensional spongy structure in the samples with a peak in the region of 60 ppm.

Also, as shown in [41], the 4-coordinated Al sites are specifically affected by the interlayer water, i.e., the peaks observed at ~60 and ~70 ppm are assigned to 4-coordinated Al which has interacted with interlayer water (~60 ppm), and that which is free from the interaction with interlayer water (~70 ppm). Based on this, it can be concluded that for samples with spherical, platy and tubular morphology of the particles, there is no interaction of 4-coordinated Al with interlayer water, and for samples sponge morphology, such an interaction is possible. Shifts in the position of four-coordinated aluminum peaks can also be associated with dehydration of the samples [41].

In ^29^Si NMR spectra of all samples, signals in the region corresponding to four-coordinated silicon, from −85 to −110 ppm, are observed [42]. It is known that ^29^Si chemical shift can provide information about considerable range of perturbation in the Si environment. The substitution by Al of each of the four silicon surrounding the central Si of a Q^4^ results in a change in the ^29^Si chemical shifts of about 5 ppm toward the negative values. Thus, the ^29^Si chemical shift range of a Q^4^ unit with no bonded Al atoms (Q^4^(0Al)) is about −102 to 116 ppm. Another peak in the spectrum of the sample with nanosponge morphology is observed in the region of −96 ppm, which indicates the presence of Q^4^(1Al) units. 

The spectra of the platy sample have two peaks at −90.5 and −91.0 ppm with almost equal heights. Both peaks are ascribed to SiO_4_ tetrahedrons with three bridged oxygen atoms and one nonbridged oxygen [43]. The bridged oxygen atoms are located between two Si atoms, whereas the nonbridged oxygen atom is located between Si and Al atoms. Notation such as Q^3^(0Al) is usually used for this type of Si. The splitting means that there are two crystallographically inequivalent Si sites. Similar splittings have been reported for kaolinite so far [44,45,46]. The spectrum of the sample with nanotubular morphology shows one broad non-split peak in the range from −94 to −90 ppm, which is most likely a superposition of two or more peaks, corresponding, as in the case of the plate sample Q^3^(0Al) units with several crystallographically inequivalent Si sites. For samples with platy and nanotubular morphology, no other peaks are observed in ^29^Si NMR spectra. For the sample with nanotubular morphology in this region, which indicates the presence of Q3(0Al) units, only a shoulder is observed, which indicates that if such units are present, then they are not in a significant amount. For the nanosponge sample, a peak is observed, which is not characteristic of any of the samples with other morphology, in the region of −82 ppm, which most likely corresponds to Q^4^(4A1) units. A sample with spherical morphology is also characterized by the presence of a broad peak in the same region; in addition, this sample also has a shoulder in the region of −85 ppm, which indicates the presence of Q^3^(1Al) units.

Thus, based on the study of samples by NMR, it can be concluded that the studied samples differ in degree of Al-for-Si substitution and the degree of the disordering of an aluminosilicate framework, which, in turn, can affect the surface properties and biological activity of the samples.

In the Appendix A the results of an additional study of samples of alumosilicates with different particle morphology are presented—by the method of FT-IR absorption spectroscopy (Appendix A) and by methods of differential thermal analysis and mass spectrometry (Appendix A).

Figure 5 shows the dependence of the surface zeta potential of the samples on pH. The zeta potential of all synthesized samples is negative in the pH range from 10 to 3.3. The zeta potential values increase with increasing pH, which coincides with the well-known trend in the zeta potential values of aluminosilicates [47,48,49,50,51]. The surface charge of aluminosilicates is largely determined by the presence of isomorphic substitutions and the breaking of Al–O–Al and Si–O–Si bonds [51]. The initial negative electrokinetic potential of the particles is largely attributed to the deprotonated silanol and aluminol groups at the pH-dependent edge faces [48]. Decreasing the pH from 10 to 3, particle surface protonation, mainly of the edge surface hydroxyl groups, occurs and, hence, the magnitude of negative electrokinetic potential of the particles decreased [52].

Raw halloysite nanotubes have a negative surface charge over the entire studied pH range, while the negative zeta potential of the surface in an alkaline medium for nanotube samples is two times higher than the negative zeta potential of the surface of samples with spherical and plate morphologies. This may be due to the fact that an external surface of halloysite is rich in siloxane (Si–O–Si) groups and an inner one comprising a gibbsite-like array of aluminol (Al–OH) groups [53]. As a result, nanotubular halloysite particles have two oppositely charged surfaces, an outer negatively charged surface and an inner positively charged surface, access to which is limited due to the small inner diameter of the tubes. A significant negative zeta potential in a wide pH range indicates a high sorption potential of particles with this morphology in relation to positively charged ions in aqueous solutions with neutral and alkaline pH values. Particles with other morphology have two outer surfaces with a negative and a positive charge. Surface force measurements revealed that the silica tetrahedral face of kaolinite is negatively charged at pH > 4, whereas the alumina octahedral face of kaolinite is positively charged at pH < 6 and negatively charged at pH > 8 [54]. As a result, samples with different morphologies have different dependences of the zeta potential on pH.

For aluminosilicates with sponge morphology, the zeta potential values in the pH range from 10 to 3.3 varies to the greatest extent (from –39.97 to +13.52 mV). The isoelectric point for the nanosponge sample is 3.2. For the rest of the samples, after extrapolation of the dependences of the zeta potential on pH to the range of low pH values, the values of the isoelectric points can be estimated as 2 (for platy and spherical particles) or less (for tubular particles).

This character of the change in the zeta potential of nanosponge samples can be associated with both the features of their morphology and their chemical composition. Thus, according to the data of chemical analysis (Table 2), the nanosponge samples are characterized by the lowest content of silicon oxide in comparison with the rest of the studied samples (43 wt %). Typically, at an acidic pH, oxide surfaces are positively charged due to protonation of surface hydroxyl groups, while at alkaline pH values, deprotonation of the terminal silanol and bridging oxygen for alumina occurs, resulting in a negative surface charge. Zeta potential curves generally fall off as pH increases. Wherein, isoelectric points for silica and aluminum are about 3.9 and 8.8, respectively. Thus, a decrease in the content of silicon oxide in a sample can lead to a shift in the position of the isoelectric point to higher pH values relative to other samples with a slightly higher content of silicon oxide. In addition, the morphological features of the nanosponge sample, which is a self-organized structure consisting of thin aluminosilicate layers and having a high specific surface area and porosity, leads to greater surface protonation and to a greater degree of leaching of the key elements (Al(III), Si(IV)) occurring at low pH values pH [48]. The features of the change in the surface zeta potential depending on pH allow us to conclude that there are significant prospects for the use of aluminosilicate particles with nanosponge morphology as universal sorbents of differently charged ions, as well as matrices of drugs that exist in aqueous solutions in the form of anions.

Equilibrium adsorption isotherms of methylene blue are shown in Figure 6. According to the Giles classification, the presented isotherms belong to the L2 type [55]. For such isotherms, the initial section is curved relative to the concentration axis, since with a decrease in the proportion of free adsorption sites, it is more difficult for adsorptive molecules to find a vacant site. The saturation limit for spherical particles is reached at an initial concentration of MB solution of 350 mg/L, and for synthetic platy, nanosponge and nanotubular particles—300 mg/L. At the same time, the adsorption capacity of the spheres increases from 9.5 to 146 mg/g with an increase in the concentration of the MB dye from 10 to 300 mg/L, for a nanosponge sample—from 8.5 to 98 mg/g, for a platy sample—from 9 to 30 mg/g, for nanotubes—from 9.2 to 57.8 mg/g. Thus, spherical particles have the highest sorption capacity for MB, while platy and tubular particles have the lowest.

The equation constants and correlation coefficients are given in Table 3. Among the three nonlinear models, the Freundlich isotherm best describes the adsorption systems of MB-plates and MB-nanotubes. The high values of the coefficients of determination (R^2^) for the Freundlich model indicate that the surface of the studied sorbents contains active centers with different affinity energies for the dye molecules. The value of 1/n can be considered as an indicator of the inhomogeneity of sorption centers: as the inhomogeneity increases, 1/n→0, and as the homogeneity of centers increases, 1/n→1. At the same time, the data obtained make it possible to characterize these aluminosilicates as materials with a high concentration of sorption centers with varying degrees of activity. The K_F_ values indicate the ease of transition of the adsorbate into the sorbent phase.

Equilibrium adsorption on spherical and nanosponge samples is most adequately described by the Temkin isotherm, the equation of which assumes the adsorbent-adsorbate chemical interaction. However, as shown by IR spectroscopy studies of samples before and after dye sorption (Appendix A), no new bands were found that could be attributed to either MB or the original sorbent. Accordingly, it can be concluded that no new chemical bonds have arisen. Therefore, the Freundlich model is more suitable for nanosponges, nanotubes and lamellar samples. For spherical particles, the Langmuir model is more appropriate, assuming a homogeneous monomolecular adsorption process. The equilibrium adsorption isotherms of methylene blue plotted according to the Langmuir model are presented in the Appendix A.

To process the kinetic data, pseudo-first-order (PFO) and pseudo-second-order (PSO) [29,30] adsorption models, as well as diffusion kinetic models [31,32] were used. The PFO and PSO models suggest that the chemical exchange reaction limits the sorption process. The criteria for assessing the accuracy and reliability of these models are the comparison of the values of the sorption capacity in the state of equilibrium q_calc_, calculated by equations with the experimental values of the sorption capacity q_exp_, as well as high values of the correlation coefficient R^2^. For all samples, the PSO model correlates well with the experimental data (Figure 7, Table 4) and indicates that chemisorption is the sorption rate-limiting step. The values of the experimental and calculated adsorption capacities are very close, which confirms the applicability of this model for describing adsorption data. The dependencies obtained allow us to conclude that the time to reach sorption equilibrium is about 2 h.

Diffusion models describe the processes of adsorbate mass transfer, which consist of the following stages: external (film) diffusion, intraparticle (gel) diffusion, and a surface reaction, which consists in the adsorption of adsorbate molecules to the inner surface of the sorbent [56]. The overall speed of the process can be controlled by one of the stages or a combination of several stages.

To assess the contribution of external and intradiffusion limitation, we used kinetic data presented in the coordinates −ln(1-F) from time t, where F is the degree of achievement of sorption equilibrium (F = q_t_/q_e_), and q_t_ from t^1/2^, where q_t_ and q_e_ are the sorption capacities (mg/g) at time t and at the moment of equilibrium.

Figure 8 shows the kinetic curves for samples with spherical, nanosponge, nanotubular and platy morphologies (Figure 8a, Figure 8b, Figure 8c and Figure 8d respectively) Kinetic curves (Figure 8a,b) for synthetic samples with spherical and sponge particle morphology in the –ln(1-F)-t coordinates are linear with a high coefficient of determination (R^2^ = 0.97 and 0.98, respectively), which may indicate an external diffusion mechanism of adsorption. In turn, for tubular and plate samples, a nonlinear dependence of the change in adsorption with time is observed in the initial part of the curve, which does not allow us to speak of a purely external diffusion control of the sorption kinetics of the dye. The tangent of the slope of the straight line to the abscissa determines the apparent rate constant of external diffusion γ, min^–1^ (Table 4).

Since the external diffusion kinetics describes only the initial stage of establishing equilibrium in the systems under consideration, in order to evaluate the contribution of the intraparticle interaction, graphic dependences of the adsorption capacity q_t_ on t^1/2^ were plotted (Figure 9). As a result of the approximation of the experimental data expressed as a linear function, two segments were identified for all samples, which indicates two stages of MB dye diffusion. The first stage corresponds to the process of external surface diffusion of the adsorbate through the solution to the external surface of the adsorbent (external diffusion mass transfer). The second stage is the diffusion of dye molecules inside the aluminosilicates to the active centers (intra-diffusion mass transfer). It can be seen from the graphs that the stage of external diffusion mass transfer (stage 1) for the samples is the fastest and lasts for 20–25 min, and then the stage of control of intraparticle diffusion (stage 2) begins, which lasts more than 75 min. In this case, the kinetic parameters corresponding to internal diffusion are characterized by the slope of the second section. The segment cut off by the continuation of this straight line on the y-axis is proportional to the thickness of the film (parameter C) surrounding the adsorbent particles. If the C parameter is not equal to zero, i.e., the straight line does not pass through the origin of coordinates, which occurs when processing the sorption kinetic data for all samples, then the contribution of the intradiffusion component decreases. At the same time, the role of “film” kinetics increases. Thus, the sorption process proceeds in the external diffusion or mixed-diffusion regions. The parameters of the intraparticle diffusion model are presented in Table 5.

The results of graphical processing of experimental data on MB adsorption using pseudo-first and pseudo-second order models and the Weber–Morris intradiffusion model show that the adsorption kinetics for all samples with a higher correlation coefficient is described using a pseudo-second order model. In addition, a comparison of the experimental and calculated adsorption values obtained using the pseudo-second order model indicates that this model gives the best agreement with the experimental results. Thus, the totality of all the results allows us to conclude that the pseudosecond order model describes the general picture of MB adsorption on the studied aluminosilicates most adequately. It should be noted that, taking into account the high values of the correlation coefficients, the kinetics of dye sorption by the samples with spherical and nanosponge particles morphology is also adequately described by the external diffusion model; therefore, it can be assumed that external diffusion makes a significant contribution to the rate of this process.

The results of the hemolytic activity study of the aluminosilicate samples with different morphologies are shown in Figure 10. Hemolysis is the process of destruction of red blood cells with the release of hemoglobin into the environment. The ease associated with isolating erythrocytes makes the hemolytic activity assay a versatile tool for rapid initial toxicity assessment [57,58]. 100% hemolysis means complete destruction of all red blood cells. From the obtained results, it is possible to draw conclusions about the effect of the particle morphology on the hemolytic activity of the samples. Thus, samples with nanotubular morphology of the particles have the highest hemolytic activity. The maximum values of hemolytic activity reach 100% at a sample concentration of 5 mg/g. Samples with platy morphology have hemolytic activity, approximately two times less than the hemolytic activity of nanotubes. Thus, the maximum hemolytic activity of platy samples is 60% at a concentration of 10 mg/g. Samples with spherical and spongy morphology have much lower hemolytic activity over the entire range of sample concentrations. Thus, at a sample concentration of 10 mg/mL, hemolysis of the spheres is 6%, which is 10 times less than that of the plates. At the same time, in the concentration range from 0 to 4%, it can be considered that the hemolytic activity of the samples is almost completely absent. If the hemolysis rate is below 5%, medical materials were considered as nonhemolysis [59]. Thus, it can be assumed that aluminosilicate nanosponges and spheres are not toxic to human blood cells, in contrast to nanotubes, which have significant hemolytic activity even at low concentrations (15% at 0.1 mg/g).

Figure 11 shows the results of a cytotoxicity study using human histiocytic lymphoma (U937) cells. The method is based on the assessment of cytotoxicity by the level of cell death induced by a particular substance. The study was carried out using a colorimetric MTT test. In this test, soluble yellow tetrazolium salt MTT (3-(4,5-dimethylthiazol-2-yl)-2,5-diphenyl-2H-tetrazolium bromide) penetrates the membrane of living cells and is reduced by mitochondrial dehydrogenases to insoluble blue formazan crystals [60]. Thus, the color intensity correlates linearly with the number of viable cells in suspension.

The results of the MTT test show that the nanotubular sample exhibits pronounced cytotoxicity. Even at the lowest sample concentration of 0.08 mg/mL, cell survival is 80%. As the sample concentration increases, the survival rate of samples sharply decreases and tends to zero at concentrations from 2.5 to 10 mg/mL Samples with other particles morphology do not have such pronounced cytotoxicity. A noticeable increase in cytotoxicity, exceeding 50 percent cell survival rate, for samples with platy morphology is observed at a sample concentration of more than 2.5 mg/mL. The cytotoxicity of nanosponge and spherical particles remains consistently low up to the concentration of 0.6 mg/mL, after which it begins to increase. Also, based on the obtained curves, the concentration value causing 50% inhibition of cell population growth (IC_50_, mg/mL) was calculated [61]. For nanotubes, this value turned out to be the smallest, 0.33 mg/mL. For samples with platy, spherical, and nanosponge morphology, IC_50_ was 1.55, 2.68, and 4.69 mg/mL, respectively.

Thus, the results of the study of hemolytic activity and cytotoxicity of the samples correlate with each other—particles with nanotubular morphology have the highest toxicity both to blood cells and tumor cells. Samples with nanosponge and spherical morphologies have the least toxicity.

The presence or absence of toxicity in inorganic compounds may be due to various factors [23]. Thus, it was shown in [62] that mesoporous silica nanoparticles show lower hemolytic activity than their nonporous counterparts of similar size. In [63] on the example of the study of seven natural clay minerals (two kaolinites, three montmorillonites, a hectorite and a palygorskite), a significant relationship between the surface area of the clays and cytotoxicity was found, however, a possible effect on the cytotoxicity of impurity phases was also noted. The surface charge of nanoparticles plays an important role in their interaction with cells and cytotoxicity [64]. The existing reports on the influence of surface charge are contradictory [65]. It can be assumed that the differences in the values of the surface zeta potential and porous-textural characteristics lead to the fact that the hemolytic activity and cytotoxicity of tubular and platy particles exceed those of particles with the morphology of nanosponges and spheres. The possible influence of the different degree of Al-for-Si substitution and the degree of the disordering of an aluminosilicate framework in the studied samples, revealed by the NMR results, cannot be ruled out.

There is a correlation between the obtained results and the results of [66]. The cytotoxicity of gold nanoparticles with different morphology was studied—rods (≈39 nm length, 18 nm width), stars (≈215 nm) and spheres (≈6.3 nm) against human fetal osteoblast, osteosarcoma and pancreatic duct cell lines by MTT and neutral-red uptake assay. AuNPs stars were found to be the most toxic and AuNPs spheres the least. Thus, the obtained results on cytotoxicity of aluminosilicate particles correlate with those available in the literature and indicate a significant effect on cytotoxicity and hemolytic activity of the particle shape.

## 4. Conclusions

Sorption properties and biological activity of aluminosilicates of the kaolinite group with different particle morphology—spherical, sponge, platy and tubular have been studied. It has been established that the studied aluminosilicate samples have the same chemical composition but differ significantly in their properties. Differences in the morphology of the nanoparticles should lead to differences in their porous-textural characteristics and specific surface values. Samples with nanosponge morphology have the highest specific surface area (470 m^2^/g). The specific surface area of the samples with spherical morphology is about 240 m^2^/g, which is also significantly (more than 10 times) higher than the values typical for the samples with platy morphology (22 m^2^/g) and 4.5 times greater than the specific surface area of nanotubular samples (55 m^2^/g). 

The sorption capacity for the methylene blue dye decreases in the series spheres-sponges-tubes-plates and is 98.2, 66.6, 38.0 and 19.5 respectively. It has been shown that the adsorption of the methylene blue on tubular, platy and nanosponge samples is best described by the Freundlich model (R^2^ = 0.97, 0.96 and 0.96, respectively), which means that adsorption on these samples proceeds by the mechanism of physical sorption. Data on dye adsorption on sample with spherical particles morphology is most adequately described by the Langmuir model, which assumes a homogeneous monomolecular adsorption process. As a result of the study of the kinetics of adsorption on synthetic aluminosilicates, it was found that sorption is well described by a pseudo-second order kinetic model, and a significant contribution of external diffusion to the rate of this process is also noted.

The results of the study of the biological activity of the samples indicate a significant effect on cytotoxicity and hemolytic activity of the particle shape. Particles with nanotubular morphology have the highest toxicity both to blood cells and tumor cells. Samples with nanosponge and spherical morphologies have the least toxicity. In conclusion, it can be noted that, according to the results of this study, aluminosilicate nanosponges and spheres can be considered as the most promising forms among the considered particles with different morphologies from the point of view of their further use in medicine, since they have a significant specific surface area, high sorption capacity and are non-toxic. 

## Figures and Tables

**Figure 1 nanomaterials-12-02559-f001:**
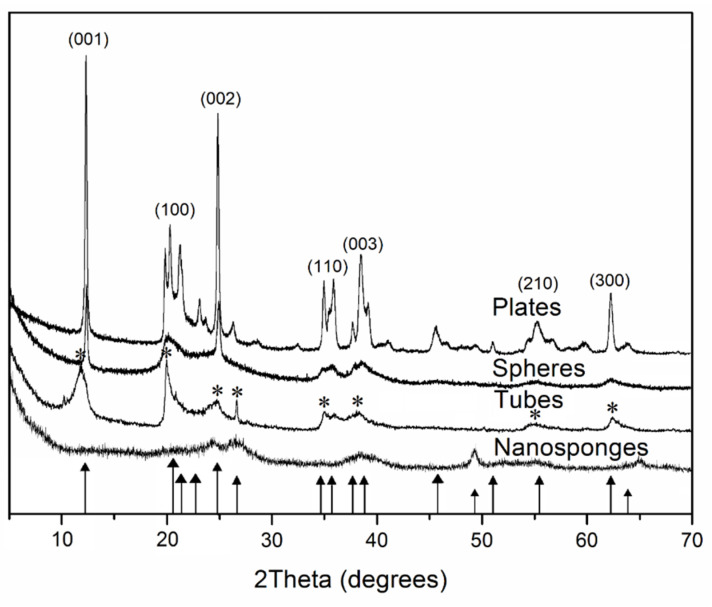
X-ray diffraction patterns of the synthetic aluminosilicate with different morphologies and raw halloysite. Bar chart of the standards: ▲—kaolinite mineral (PDF No. 79-1593); *—raw halloysite (PDF No 00–009-0453).

**Figure 2 nanomaterials-12-02559-f002:**
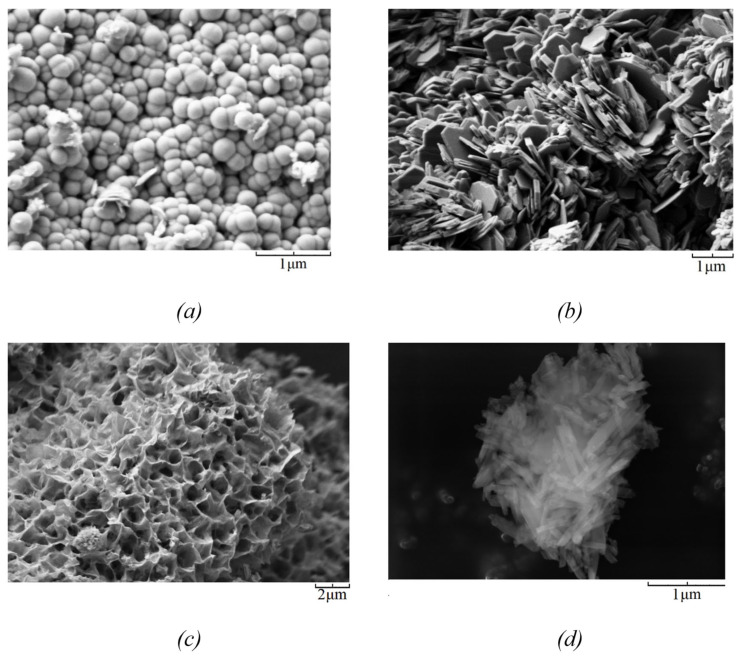
SEM images of the samples with different morphology: (**a**)—spherical; (**b**)—platy; (**c**)—sponge; (**d**)—nanotubular.

**Figure 3 nanomaterials-12-02559-f003:**
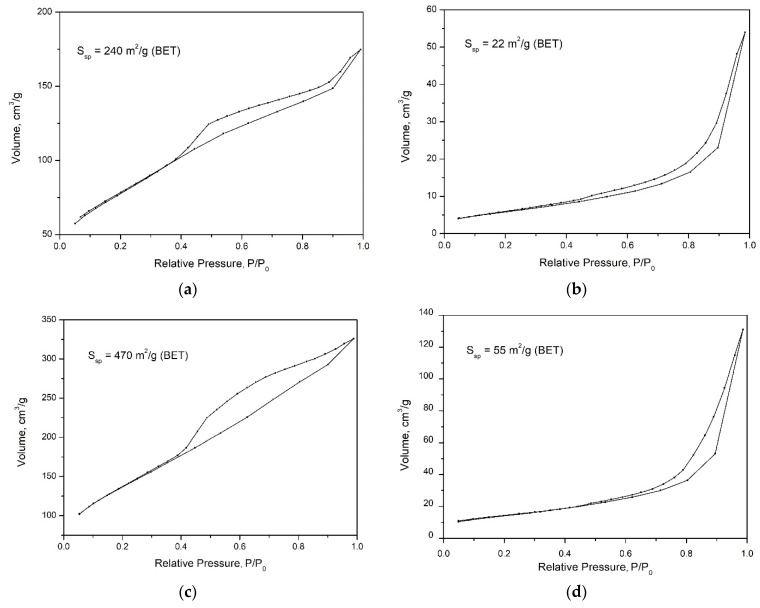
Nitrogen sorption–desorption isotherms of aluminosilicates of different morphology: (**a**)—spheres, (**b**)—plates, (**c**)—sponges, (**d**)—nanotubes.

**Figure 4 nanomaterials-12-02559-f004:**
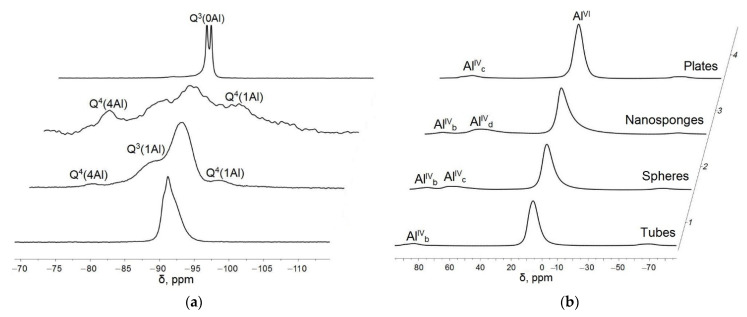
NMR spectrum: (**a**)—^29^Si; (**b**)*—*^27^Al.

**Figure 5 nanomaterials-12-02559-f005:**
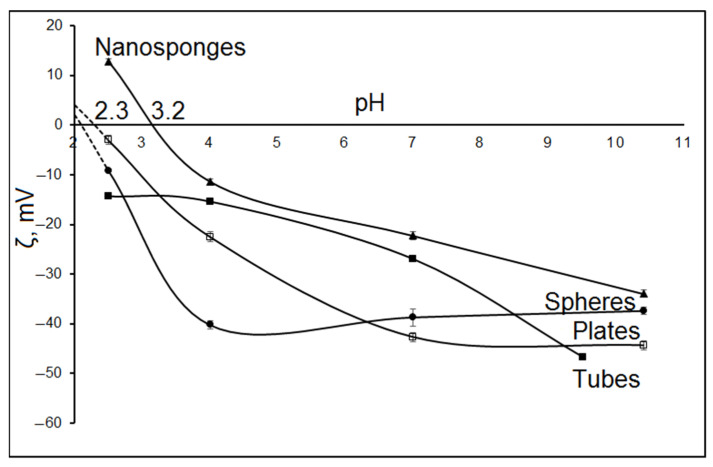
Dependence of the zeta potential of the sample surface on pH.

**Figure 6 nanomaterials-12-02559-f006:**
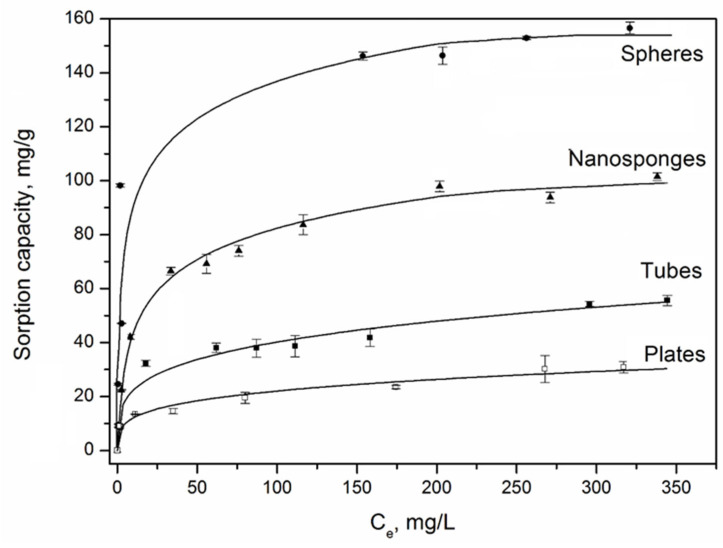
The isotherms of MB sorption by samples with different morphology: ●—spheres; ▲—nanosponges (Temkin model), ■—nanotubes, □—plates (Freundlich model).

**Figure 7 nanomaterials-12-02559-f007:**
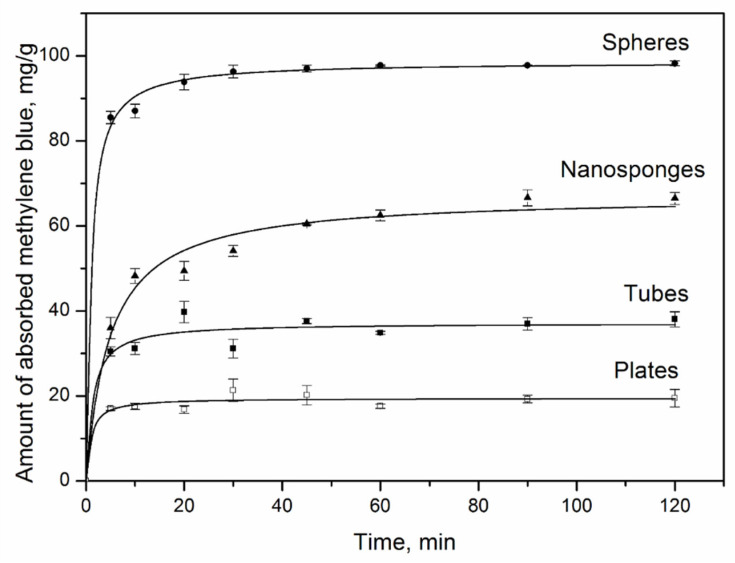
The integral kinetic curves of PSO sorption of MB by samples with different morphologies: ●—spheres, ▲—nanosponges, ■—nanotubes, □—plates.

**Figure 8 nanomaterials-12-02559-f008:**
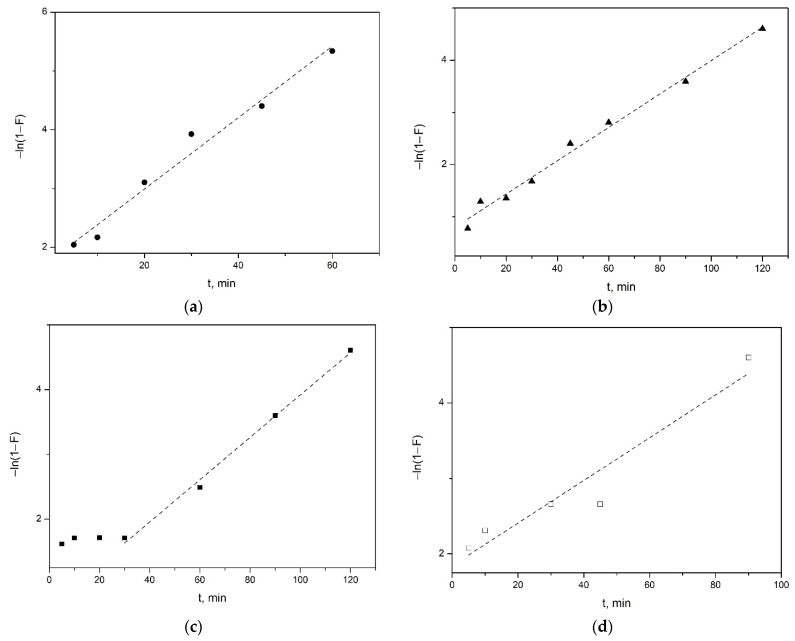
Dependence “−ln(1-F)-t” in systems: “MB—porous aluminosilicate” for the samples with different morphology: (**a**)—spheres, (**b**)—nanosponges, (**c**)—nanotubes, (**d**)—plates.

**Figure 9 nanomaterials-12-02559-f009:**
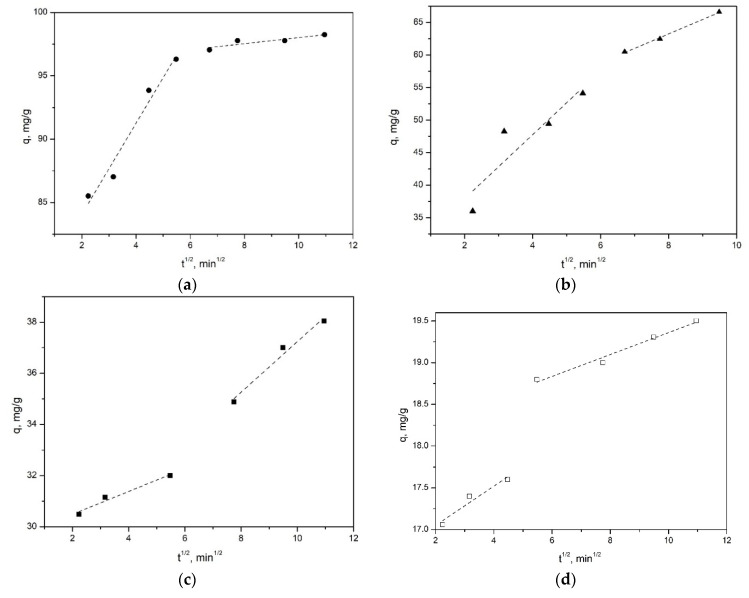
The dependence of q_t_ on t during sorption of MB by the samples with different morphologies: (**a**)—spheres, (**b**)—nanosponges, (**c**)—nanotubes, (**d**)—plates.

**Figure 10 nanomaterials-12-02559-f010:**
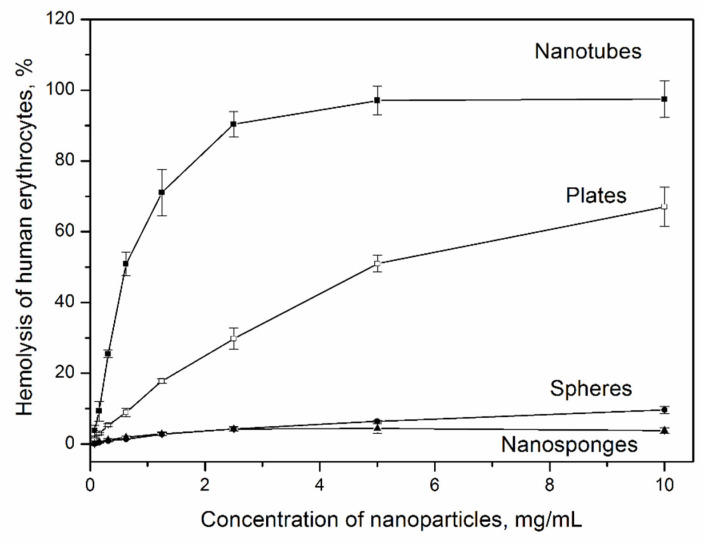
The hemolytic activity of the particles with different morphology.

**Figure 11 nanomaterials-12-02559-f011:**
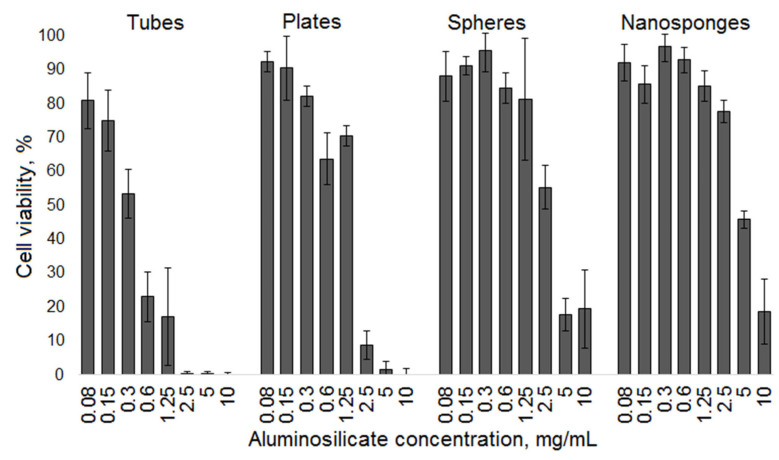
The effect of samples morphology on cell viability according to MTT test results.

**Table 1 nanomaterials-12-02559-t001:** Samples morphology and synthesis conditions.

Particles Morphology	Synthesis Conditions		
	Temperature, °C	Time, Days	pH
spheres	220	3	12
plates	350	4	7
sponges	220	3	2.6
nanotubes *	-	-	-

* raw halloysite nanotubes (Sigma-Aldrich).

**Table 2 nanomaterials-12-02559-t002:** The results of a chemical analysis of the samples.

Chemical Formula (by Synthesis)	Particles Morphology	Chemical Composition, wt. %		
		SiO_2_	Al_2_O_3_	Loss on Ignition
Al_2_Si_2_O_5_(OH)_2_	spheres	44.74	37.22	14.74
Al_2_Si_2_O_5_(OH)_2_	plates	45.84	39.48	14.05
Al_2_Si_2_O_5_(OH)_2_	sponges	43.77	36.14	15.79
Al_2_Si_2_O_5_(OH)_2_·nH_2_O	tubes	46.22	36.38	16.04

**Table 3 nanomaterials-12-02559-t003:** Equation constants of MB sorption isotherms.

Samples Morphology	q_exp_	Langmuir Equation			Freundlich Equation			Temkin Equation		
		q_m_	K_L_	R^2^	n	K_F_	R^2^	B_T_	A_T_	R^2^
Nanotubes	55.60	48.80	0.09	0.90	3.94	12.50	0.97	6.94	4.34	0.96
Spheres	146.0	147.1	0.41	0.87	4.38	45.15	0.83	20.07	9.12	0.90
Nanosponges	98.2	91.5	0.09	0.96	3.16	18.94	0.96	16.76	1.36	0.99
Plates	31.0	30.66	0.04	0.81	3.82	6.63	0.96	3.68	5.28	0.89

q_m_—maximum sorption capacity (mg/g), q_exp_—experimental value of sorption capacity (mg/g); K_L_—Langmuir constant related to adsorption free energy (L/g); B_T_—constant related to the heat of adsorption (L/g); K_F_—Freundlich constant related to adsorbent capacity; A_T_—dimensionless Temkin isotherm constant.

**Table 4 nanomaterials-12-02559-t004:** The parameters of kinetic models of sorption of MB on aluminosilicates with different morphologies.

Samples Morphology	q_exp_, mg/g	PFO Model			PSO Model		
		q_calc_	k_1_	R^2^	q_calc_	k_1_	R^2^
Nanotubes	38.0 ± 1.7	36.00	0.33	0.94	37.19	0.02	0.95
Spheres	98.2 ± 0.6	95.9	0.41	0.98	98.59	0.01	0.99
Nanosponges	66.6 ± 1.4	62.41	0.126	0.95	67.16	0.003	0.98
Plates	19.5 ± 2.0	19.00	0.43	0.94	19.5	0.06	0.95

**Table 5 nanomaterials-12-02559-t005:** The parameters of sorption diffusion models of on the samples with different morphologies.

Samples Morphology	External Diffusion Model		Intraparticle Diffusion Model			
	γ·10^−2^, min^–1^	R^2^	K_d_, mg/(g·min^0.5^)	C	R^2^ (Stage 1)	R^2^ (Stage 2)
Nanotubes	3.27	0.99 (without curve start)	0.99	27.32	0.93	0.95
Spheres	6.01	0.97	0.23	95.60	0.95	0.81
Nanosponges	3.19	0.98	0.98	45.48	0.82	0.99
Plates	2.82	0.90	0.23	16.57	0.88	0.97

## Data Availability

Not applicable.

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
