# Peer review of "Particles Morphology Impact on Cytotoxicity, Hemolytic Activity and Sorption Properties of Porous Aluminosilicates of Kaolinite Group"

_nanomaterials, 2022, doi:10.3390/nano12152559_

Round 1

Reviewer 1 Report

This work deals with a comparison of aluminosilicate materials of the kaolinite group with different particle morphologies. The topic and the completed analyses have significant originality and the manuscript merits publication in the journal Nanomaterials. The paper can be accepted, subject to minor revision (including the correction of some small typos).

I have only one conceptual comment. The X-ray diffractions patterns and the TG/DSC curves (and, to some extent, the IR spectra) suggest that the aluminosilicate nanosponges and spherical particles are mostly amorphous gel-like materials. The characteristic X-ray peak around 0.7 nm is even absent for the nanosponges. Therefore, it should be emphasized in the paper that these materials, even if they have the same chemical composition, are not necessarily similar crystalline aluminosilicates (of the kaolinite subgroup) to their platy and tubular counterparts. These structural (and not just morphological) differences can also lead to differences in the studied properties. 

Minor comments

The abbreviation 'MG' is frequently used in the text instead of 'MB' (methylene blue).

Kaolin rock and kaolinite mineral should be distinguished in the text.

Line 18, page 10: Use '...while at alkaline pH values...' instead of '...while at acidic pH values...'.

Line 2, page 12: I recomment to use '...could be attributed to...' instead of '...were not characteristic of...'.

Some presumably intended superscripts on page S4 should be corrected in the SUPPORTING INFORMATION. 

Author Response

Response to Reviewer 1 Comments

The authors would like to thank the Reviewer for valuable remarks and comments, which the authors have tried to take into account or comment on.

Point 1. I have only one conceptual comment. The X-ray diffractions patterns and the TG/DSC curves (and, to some extent, the IR spectra) suggest that the aluminosilicate nanosponges and spherical particles are mostly amorphous gel-like materials. The characteristic X-ray peak around 0.7 nm is even absent for the nanosponges. Therefore, it should be emphasized in the paper that these materials, even if they have the same chemical composition, are not necessarily similar crystalline aluminosilicates (of the kaolinite subgroup) to their platy and tubular counterparts. These structural (and not just morphological) differences can also lead to differences in the studied properties.

Response 1. The authors are confident that samples with spherical and nanosponge morphology cannot be considered as amorphous, despite some blurring of the X-ray diffraction pattern. The X-ray diffraction pattern of such polytypes essentially depends on the degree of perfection of the crystal structure. An amorphous state is a condensed state of substances whose atomic structure has a short-range order and no long-range order. Amorphous substances do not form crystalline faces and do not show differences in properties in different directions. As shown by our numerous studies of the samples with nanosponge and spherical morphologies (for an example, O.Y. Golubeva et al. Influence of hydrothermal synthesis conditions on the morphology and sorption properties of porous aluminosilicates with kaolinite and halloysite structures, Appl. Clay Sci. 199 (2020). https://doi.org/10.1016/j.clay.2020.105879; O.Y. Golubeva et al. Aluminosilicate Nanosponges: Synthesis, Properties, and Application Prospects., Inorg. Chem. 60 (2021) 17008–17018. https://doi.org/10.1021/acs.inorgchem.1c02122) using electron microscopy, in particular, using focused ion beam-scanning microscopy (FIB-SEM), such particles have crystalline faces. In particular, on the cuts of nanosponge samples, it is clearly seen that aluminosilicate layers are formed, which are characteristic of the kaolinite structure, which self-organize with the formation of a nanosponge structure. In this way, they have both short-range and long-range order. In this case, the imperfection of the crystal structure leaves an imprint on the X-ray diffraction pattern. The X-ray diffraction pattern of spherical samples is absolutely typical for polytypes of the kaolinite subgroup. Besides, our previous studies (O.Y. Golubeva et al. Influence of hydrothermal synthesis conditions on the morphology and sorption properties of porous aluminosilicates with kaolinite and halloysite structures, Appl. Clay Sci. 199 (2020). https://doi.org/10.1016/j.clay.2020.105879) of the influence of the synthesis conditions (temperature and duration of synthesis) showed that spherical particles with an average diameter of 50 nm are formed on the surface of layered particles. As the duration of synthesis increases, the spherical particles grow and their average diameter reaches 300–500 nm. In addition, spherical particles are not destroyed by ultrasound. These facts indicate the crystal structure of the samples.

Responses to minor comments

Point 1. The abbreviation 'MG' is frequently used in the text instead of 'MB' (methylene blue).

Response 1. Thank you very much for your comment. The authors took it into account and made the appropriate corrections.

Point 2. Kaolin rock and kaolinite mineral should be distinguished in the text.

Response 2. Thank you very much for this note. The authors have made appropriate clarifications (line 163, 187).

Point 3. Line 18, page 10: Use '...while at alkaline pH values...' instead of '...while at acidic pH values...'.

Response 3. The authors have made the appropriate corrections (p. 10, line 313).

Point 4. Line 2, page 12: I recomment to use '...could be attributed to...' instead of '...were not characteristic of...'.

Response 4. The authors have made the appropriate corrections (p.12, line 359).

Point 5. Some presumably intended superscripts on page S4 should be corrected in the SUPPORTING INFORMATION.

Response 5. The appropriate corrections have been made to the text of the Supporting information file (S4, S6).

Reviewer 2 Report

This is an interesting and comprehensive study which attempts to reveal the role of the morphology of Al-Si nanoparticles on their adsorptive parameters and cytotoxicity.

The study is performed rather accurately, and can be recommended for publication in Nanomaterials after some corrections.

Major suggestions.

1. In addition to Figure 6, or instead of it, it would be instructive to see a graph with all isotherm data and models of only one isotherm, the Langmuir one, in spite of its lower R2 factors. This would show more reliably the sorption capacities of the materials.

2. Since there is an obvious and expected dependence of MB adsorption capacity on the specific surface, a correlation plot could be constructed to show this relation for the differently shaped particles.

Most importantly, authors should discuss, whether the particle shape, or the specific surface is the main factor of the observed differences of the various characteristics.

Minor suggestions

3. Figure 10 shows very large error bars for the nanotube sample. Given the rather monotonic behavior of the data at low concentrations, it can be thought that the error bars are overestimated. Authors may re-check their evaluation or comment this remark.

4. In some references the page number/article number is missing, e.g. ref.11.

Author Response

Response to Reviewer 2 Comments

The authors would like to thank the Reviewer for a careful reading of the manuscript and for valuable comments and remarks, which the authors tried to take into account and make the appropriate changes and additions.

Major suggestions

Point 1. In addition to Figure 6, or instead of it, it would be instructive to see a graph with all isotherm data and models of only one isotherm, the Langmuir one, in spite of its lower R2 factors. This would show more reliably the sorption capacities of the materials.

Response 1. In accordance with the proposals of the Reviewer, the authors prepared an additional figure with equilibrium adsorption isotherms of methylene blue plotted according to the Langmuir model. Figure S5 has been placed in the Supporting Information file. The corresponding link to the figure is given in the text of the manuscript.

Point 2. Since there is an obvious and expected dependence of MB adsorption capacity on the specific surface, a correlation plot could be constructed to show this relation for the differently shaped particles. Most importantly, authors should discuss, whether the particle shape, or the specific surface is the main factor of the observed differences of the various characteristics.

Response 2. An unambiguous correlation between the sorption capacity of samples with different particle morphology in relation to methylene blue with their specific surface was not found. Thus, nanosponges have the largest specific surface area (470 m2/g), however, their sorption capacity is almost two times lower than that of particles with a spherical morphology, the specific surface area of which is about 240 m2/g. Therefore, a correlation plot cannot be constructed. From the results obtained, it follows that the sorption capacity, as well as hemolytic activity and cytotoxicity, is influenced by a complex of factors - specific surface area, particles shape, surface charge, and distribution of active sites on the surface.

Minor suggestions

Point 3. Figure 10 shows very large error bars for the nanotube sample. Given the rather monotonic behavior of the data at low concentrations, it can be thought that the error bars are overestimated. Authors may re-check their evaluation or comment this remark.

Response 3. The authors agree with the Reviewer's remark that the error bars for a nanotubular sample are very large (Fig. 10, Hemolytic activity of the samples). Realizing this, the authors additionally carried out a number of additional measurements. The magnitude of the error has decreased somewhat, but not significantly. In the corrected version of the manuscript, we present Fig.10 with new data. At the same time, the character of the dependence has not changed. A large error can be associated with the structural peculiarities of tubular samples, which consist in the presence of two differently charged surfaces inside and outside the tubes. There is some possibility that the destruction of erythrocytes occurs differently depending on which surface - external or internal - they are in contact with, which, in turn, can lead to a significant measurement error.

Point 4.  In some references the page number/article number is missing, e.g. ref.11.

Response 4. The authors thank the Reviewer for this remark. Corresponding changes have been made to the bibliography.

Round 2

Reviewer 2 Report

Authors have addressed all points raised by the Reviewer in a satisfactory way.